# ProRA: Projection Aware Low-Rank Adaptation for Parameter Efficient Fine-Tuning

## Abstract

Despite the remarkable success of large language models (LLMs) across diverse tasks, the computational cost of fine-tuning them remains high. Low-Rank Adaptation (LoRA) addresses this by updating through the product of two low rank matrices. LoRA initializes low-rank matrices using random Gaussian noise and zeros, while keeping the pretrained weights frozen. However, such random and zero initialization leads to slow convergence and limits expressiveness. To overcome these limitations, we propose **Pro**jection Aware Low-**R**ank **A**daptation (ProRA). ProRA initializes adapter matrices by projecting the original weight matrix into its orthonormal subspace and keeps the residual weight matrix frozen. ProRA leverages the orthonormal projection to ensure that updates preserve the geometric structure of pretrained models and are aligned with orthogonal subspaces, leading to faster convergence and improved performance. Furthermore, we interpret ProRA through the lens of geometric complexity. ProRA lowers geometric complexity in the frozen weights, which facilitates more efficient fine-tuning. Our proposed ProRA demonstrates empirical superiority over LoRA across diverse tasks. On the GSM8K benchmark dataset, ProRA achieves 78.11% accuracy with GEMMA-7B, outperforming LoRA's 74.53% by 3.58%. Comparative evaluations across various model architectures consistently show that ProRA outperforms LoRA, highlighting its robustness and effective fine-tuning capability.

## 1 Introduction

Large Language Models (LLMs) are at the forefront of progress in Natural Language Processing (NLP) (Hosseini and Fedorenko, 2023; Zheng et al., 2023; Creswell et al., 2023; Yu et al., 2024; Luo et al., 2024). Their success can largely be attributed to transfer learning (Strangmann et al., 2024; Wang et al., 2024; Raffel et al., 2020). Among the various transfer learning strategies, the most widely adopted approach involves two key stages. The first stage, known as pretraining, involves training the LLMs on large-scale, general-purpose datasets using either supervised or unsupervised learning objectives. The subsequent stage, known as fine-tuning, focuses on adapting the pretrained model to a specific downstream task by updating its weights (Bengio, 2012). Generally these downstream tasks are unknown at the time of pre-training. LLMs require fine-tuning to achieve optimal performance on downstream tasks. While fine-tuning is highly effective for adapting LLMs to task-specific datasets, the process is computationally intensive, requiring significant time and memory resources. To overcome these challenges, various Parameter-Efficient Fine-Tuning (PEFT) techniques have been proposed (Houlsby et al., 2019; Zhang et al., 2023; Liu et al., 2024b). These PEFT methods focus on updating only a small subset of the parameters to achieve efficient adaptation. PEFT includes a variety of techniques, such as tuning only select layers (partial fine-tuning) (Zaken et al., 2022; Lawton et al., 2023; Sung et al., 2021; Xu et al., 2021), using learnable input embeddings (soft prompts) (Hambardzumyan et al., 2021; Wang et al., 2023), and applying low-rank matrix factorization during adaptation (Kopiczko et al., 2024; Hu et al., 2022; Zhang et al., 2023; Aghajanyan et al., 2021). Among these methods, Low-Rank Adaptation (LoRA) (Hu et al., 2022) is notable for using two low-rank matrices to approximate parameter updates. LoRA achieves comparable performance to full fine-tuning with significantly fewer trainable parameters.

LoRA and its successors operate on the hypothesis that parameter adaptation can be effectively achieved using low-rank matrices. In LoRA, the pre-trained weights are updated through the product of two low-rank matrices. These two low-rank matrices are initialized such that one follows a

random Gaussian distribution, and the other is set to zero (Hu et al., 2022; Hayou et al., 2024). As a result, their initial product is a zero matrix, ensuring no alteration to the model's output at initial step. LoRA eliminates the need to compute gradients or maintain optimizer states for the original weight matrix by optimizing two low-rank matrices instead. This approach reduces the number of trainable parameters by up to 10,000 times and significantly reduces memory requirements (Hu et al., 2022).

Furthermore, low-rank adaptation techniques can be analyzed through the lens of geometric complexity. Geometric complexity quantifies the variability of the function learned by a model (Dherin et al., 2022). Recent work (Munn et al., 2024), establishes a theoretical relationship between the geometric complexity of a pretrained model and its fine-tuning performance. Notably, the geometric complexity of a pretrained network directly influences its effectiveness in transfer learning. Models with lower geometric complexity tend to exhibit better generalization, leading to improved transfer accuracy during fine-tuning (Munn et al., 2024).

Despite having several benefits, we identify two key challenges of LoRA.

- **Slower Convergence:** Unlike full fine-tuning, LoRA initially preserves the output of the pretrained model for a given input, as the pretrained weights remain frozen. Consequently, the updates to the model's output rely entirely on the product of two low-rank matrices. Since these matrices are typically initialized with Gaussian noise and zeros, these updates causing gradients remain informative for a longer duration during the initial steps, and this initialization leads to slower convergence during the early stages of fine-tuning.

- **High geometric complexity:** LLMs possess significantly higher geometric complexity, which arises from both their architectural design and training processes that emphasize expressive power over simplicity (Valeriani et al., 2023; Munn et al., 2024; Cosentino and Shekkizhar, 2024). Higher geometric complexity hinders effective adaptation during fine-tuning.

To address these challenges, we propose a unified approach called **Projection Aware Low-Rank Adaptation (ProRA)**, effectively tackling both issues with a single solution. ProRA separates the pretrained weights into a trainable low-rank projection and a frozen residual component. By initializing trainable parameters by the projection using orthonormal subspaces, enables faster convergence. Also with preserving the residual with lower geometric complexity, improves transfer learning performance empirically.

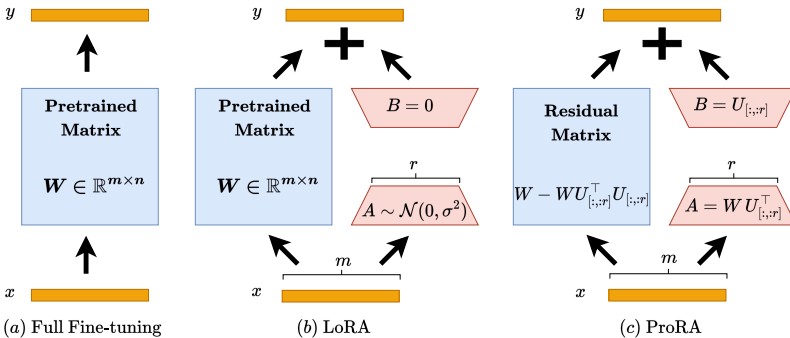

Figure 1: Comparison of Full Fine-tuning, LoRA, and ProRA approaches. In this illustration, blue modules denote frozen parameters during training, while pink modules highlight components that are updated.

In ProRA, adapters are derived directly from the pretrained weight matrix $W$ by decomposing it into two distinct components: a projection matrix $W_{\text{proj}}$ and a residual matrix $W_{\text{res}}$. The projection matrix $W_{\text{proj}}$ obtained by projecting $W$ onto a low dimensional orthonormal subspace $U^{\top}$. The residual matrix $W_{\text{res}}$, which represents the difference between the original matrix $W$ and its projection $W_{\text{proj}}$. The projection matrix $W_{\text{proj}}$ is expressed as the product of two low-rank matrices, $A$ and $B$, both of which are trainable. To obtain this initialization, the pretrained matrix $W$ is first compressed by

projecting it onto a low-dimensional orthonormal subspace, yielding matrix $A$. Subsequently, $B$ is then initialized as an orthonormal matrix, thereby reconstructing a low-rank approximation of $W$. Since $A$ is derived through orthonormal projection, it inherently preserves the Frobenius norm of the original weight matrix. $B$, with its orthonormal structure, exhibits favorable geometric properties, which can lead to a better-conditioned optimization landscape (Huang et al., 2018). Leveraging both norm preservation and structured orthonormal subspaces leads to faster convergence and in an appropriate direction. The proposed ProRA approach also aims to keep the geometric complexity of frozen weights lower. $W_{\text{res}}$ of the pretrained matrix is utilized to reduce the geometric complexity of the frozen component. In this paper, we have theoretically shown that the frozen weights in ProRA have lower geometric complexity than the pretrained weight $W$. This lower geometric complexity of frozen weights allows for better transfer learning, resulting in better performance empirically. A comparison between full fine-tuning, LoRA, and the proposed ProRA is shown in Figure 1.

This paper makes the following key contributions:

- We introduce Projection Aware Low-Rank Adaptation (ProRA), a unified framework that performs low-rank adaptation along orthonormal directions while minimizing geometric complexity.

- We propose a novel initialization method named ProRA. By initializing trainable parameters through projections onto orthonormal subspaces, ProRA enables stable gradient flow and better-conditioned updates. This structured initialization significantly accelerates convergence.

- ProRA enhances transfer learning performance by reducing the geometric complexity of the frozen residual weights during fine-tuning. Lower geometric complexity allows for better generalization to downstream tasks.

- We demonstrate both theoretically and empirically that ProRA maintains lower geometric complexity in the frozen components, leading to improved performance and accelerated convergence.

## 2 PRELIMINARY

### 2.1 LoRA

LoRA is a prominent contribution in the area of PEFT. It freezes the weights of pretrained models and integrates trainable low-rank matrices into each layer of the transformer. Given a pretrained weight matrix $W$, LoRA approximates the weight update using a low-rank decomposition:

$$\Delta W = AB,$$

where $A \in \mathbb{R}^{p \times r}$, $B \in \mathbb{R}^{r \times q}$, and the rank $r \ll \min(p, q)$. The modified forward pass is given by:

$$Y = (W + \Delta W)X.$$

Matrix $A$ and $B$ are trainable parameter, while pretrained weights $W$ are frozen during finetuning. Trainable parameters are initialized with Gaussian noise and zero matrix of appropriate dimension.

**LoRA varients:** In recent years, following the introduction of LoRA, several variants have been proposed to further improve parameter efficiency. VERA (Vector-based Random Matrix Adaptation) (Kopiczko et al., 2023) reduces the number of trainable parameters by employing two diagonal matrices that are shared across layers. AdaLoRA (Zhang et al., 2023) dynamically learns the optimal rank for each layer during training. Another approach, DoRA (Decomposed Low-Rank Adaptation) (Liu et al., 2024a), factorizes the parameter matrix into directional and magnitude components and applies low-rank adaptation to reduce trainable parameters. An alternative method, PiSSA (Principal Singular Value and Singular Vector Adaptation) (Meng et al., 2024), initializes the low-rank matrices $B$ and $A$ using the principal singular vectors and singular values of the pretrained weight matrix, enabling faster convergence. Moreover, OLoRA (Büyükakyüz, 2024) leverages the orthonormal decomposition of pretrained weight matrices to initialize low-rank adapters. Similarly, SVFT (Lingam et al., 2024) is another PEFT technique that utilise adaptation singular vector decomposition of pretrained weight matrix.

## 2.2 GEOMETRIC COMPLEXITY ((DHERIN ET AL., 2022))

For a model function $f : \mathbb{R}^d \to \mathbb{R}^p$, the **geometric complexity** is quantified as:

$$\text{GC}(f) = \mathbb{E}_{x \sim \mathcal{Q}} \left[ \|\nabla_x f(x)\|_F^2 \right], \tag{1}$$

where $\nabla_x f(x)$ represents the Jacobian of $f$ with respect to the input $x$, $\|\cdot\|_F$ is the Frobenius norm, and the expectation is taken over a data distribution $\mathcal{Q}$.

**Geometric Complexity in Linear Mappings:** For a linear transformation $f(x) = Wx + b$, where $W \in \mathbb{R}^{p \times d}$ is the weight matrix, the geometric complexity becomes:

$$\text{GC}(f) = \|W\|_F^2. \tag{2}$$

Here, $\|W\|_F^2 = \sum_{i=1}^p \sum_{j=1}^d W_{i,j}^2$ represents the squared Frobenius norm of the weight matrix. This expression aligns with the concept of discrete Dirichlet energy.

## 3 PRORA: PROJECTION AWARE LOW-RANK ADAPTATION

In this section, we formally introduce our proposed method, **Projection Aware Low-Rank Adaptation (ProRA)**, for fine-tuning pretrained LLMs. The central innovation of ProRA lies in controlled geometric complexity and its geometry aware update strategy. Unlike traditional low-rank adaptation approaches, ProRA begins by considering the entire set of pretrained weights, denoted as $W$. ProRA decomposes the pretrained weights $W$ into two matrices such that one is orthonormal to the other. This decomposition is motivated by the fact that orthonormal matrices exhibit favorable geometric properties, leading to a better-conditioned optimization landscape. Specifically, the weights are split into a *residual part*, $W_{\text{res}}$, which is kept frozen during training, and a *projected part*, $W_{\text{proj}}$, which contains trainable parameters. We first focus on the residual part so that it has lower geometric complexity. The residual component preserves geometric structures because it remains orthogonal to the subspace spanned by the original weights, while the projected component is obtained by projecting the original matrix onto a low rank matrix. The *projected part*, $W_{\text{proj}}$ is further decomposed into two low rank matrices.

Mathematically, this can be expressed as

$$W = W_{res} + W_{proj}. \tag{3}$$

Where $W_{proj} = WP = WU_{[:,:r]}^T U_{[:,:r]}$ and $U_{[:,:r]}$ is an orthonormal subspace of $W$. $U_{[:,:r]}$ is obtained by taking top $r$ column of orthonormal subspace of $W$. The residual is given by

$$W_{res} = W - WU_{[:,:r]}^T U_{[:,:r]}. \tag{4}$$

Since $W_{\text{res}}$ is frozen, only $W_{\text{proj}}$ is updated during fine-tuning. To ensure compatibility with the LoRA architecture, $W_{\text{proj}}$ is further decomposed into two low-rank matrices,

$$A = WU_{[:,:r]}^T \in \mathbb{R}^{p \times r}, \tag{5}$$

and

$$B = U_{[:,:r]} \in \mathbb{R}^{r \times q}. \tag{6}$$

Hence $W_{proj} = AB$, and both $A$ and $B$ are low rank matrices with or lower rank than $r$. Consequently, the output of the layer can be written as

$$Y = WX = (W_{res} + W_{proj}) = (W_{\text{res}} + AB)X, \tag{7}$$

which maintains full compatibility with the pretrained model during fine-tuning. The gradients of $B$ and $A$ are given by

$$\frac{\partial \mathcal{L}}{\partial B} = A^\top \frac{\partial \mathcal{L}}{\partial Y} X^\top, \quad \text{and} \quad \frac{\partial \mathcal{L}}{\partial A} = \frac{\partial \mathcal{L}}{\partial Y} X^\top B^\top,$$

respectively. Here, $\frac{\partial \mathcal{L}}{\partial Y}$ denotes the gradient of the loss with respect to the layer output $Y$. Since $U$ is initialized as an orthonormal matrix, $A$ consequently preserves the Frobenius norm. The orthonormal initialization of $B = U$ ensures that adaptation occurs within a well-conditioned subspace, enabling ProRA to converge more quickly. In contrast, LoRA initializes its $A$ and $B$ matrices with Gaussian noise and zero adapters in the early stages, potentially wasting initial gradient descent steps. Such uninformative initialization in LoRA can lead to suboptimal local solutions and degraded performance.

Since ProRA ultimately reduces to the LoRA architecture (equation 7), it inherits most of LoRA's benefits, including a reduced number of trainable parameters. Furthermore, we provide a theoretical analysis of ProRA's adaptation properties, demonstrating its advantages in terms of convergence speed, parameter efficiency, and preservation of geometric structure within the model's weights.

### 3.1 GEOMETRIC COMPLEXITY OF $W_{res}$

While constructing the ProRA approach, we take into account that the geometric complexity of the frozen model becomes lower. Various studies have shown that lower geometric complexity is responsible for better transfer of knowledge from pretrained weights and improved fine-tuning performance (Dherin et al., 2022). In the literature, low-rank adaptation techniques for fine-tuning have not been explored through the lens of geometric complexity using the Dirichlet energy function. To ensure lower geometric complexity, we split the original weight matrix such that the frozen and trainable parts are orthonormal to each other. In this way, we are able to explicitly control geometric complexity. Our proposed ProRA framework is specifically designed to keep the geometric complexity of the frozen weights low, while updating only a small subset of the pretrained weights. The architecture of ProRA, which is similar to LoRA, allows us to control the geometric complexity of the frozen weights. In the forward pass, the change in output $Y$ is achieved by updating the weights linearly via the equation $Y = (W + \Delta W)X$. Thus, utilizing linear weight updates, we can express geometric complexity in terms of the Frobenius norm of the weight matrix. We have theoretically shown that the geometric complexity of the frozen part is lower than that of $W$, i.e., the frozen part in LoRA. Empirical results also show that explicitly controlling geometric complexity through orthonormal splitting leads to faster convergence and better performance compared to both PiSSA and LoRA.

### 3.2 THEORITICAL PROPERTIES OF PRORA

**Theorem 1** (Orthogonality of Residual and Projected Weights at Initialization). *The residual component $W_{res}$ and the projected component $W_{proj}$ are orthogonal under the Frobenius inner product, i.e.*

$$\langle W_{res}, W_{proj} \rangle_F = \text{tr}(W_{res}^\top W_{proj}) = 0.$$

*Proof.* Let $P = U_{[:,:r]}^\top U_{[:,:r]}$ denote the orthonormal projection matrix onto the column space of $U$. By construction, $P$ is idempotent ($P^2 = P$) and symmetric ($P^\top = P$). The projected and residual components can be expressed as

$$W_{\text{proj}} = WP, \quad \text{and} \quad W_{\text{res}} = W(I - P),$$

where $I$ is the $k \times k$ identity matrix.

The Frobenius inner product between $W_{\text{res}}$ and $W_{\text{proj}}$ is

$$\text{tr}(W_{\text{res}}^\top W_{\text{proj}}) = \text{tr}\left((W(I - P))^\top (WP)\right)$$
$$= \text{tr}\left((I - P)W^\top WP\right)$$
$$= \text{tr}\left(W^\top WP(I - P)\right).$$

Since $P(I - P) = P - P^2 = P - P = 0$, it follows that

$$\text{tr}(W_{\text{res}}^\top W_{\text{proj}}) = \text{tr}(W^\top W \cdot 0) = 0.$$

$\square$

**Theorem 2.** *The residual matrix $W_{res}$ has lower geometric complexity than the original pretrained weight matrix $W$.*

*Proof.* Proof of this theorem is described in Appendix A.1. □

### 3.3 INTERPRETATION OF $W_{proj}$

We interpret $W_{\text{proj}}$ in terms of the down-projection and reconstruction of the pretrained weights $W$. In the first initialization step, ProRA compresses the pretrained weights by projecting them onto a low-rank orthonormal submatrix, given by $A = W U_{[:,:r]}^{\top}$. The second part of $W_{\text{proj}}$ is $B = U_{[:,:r]}$. After down-projection using $U_{[:,:r]}^{\top}$, we reconstruct $W$ via an up-projection using $U_{[:,:r]}$. Since orthonormal matrices belong to the orthogonal group, they provide favorable geometric conditions for optimization (Huang et al., 2018). Hence, initializing $A$ and $B$ in this way preserves the Frobenius norm and yields a well-conditioned optimization landscape for adaptation in ProRA.

### 3.4 COMPARATIVE ANALYSIS OF PRORA AND LORA VARIANTS: AN INITIALIZATION VIEW

In this subsection, we present a comparison between our proposed method and LoRA along with its variants. While many successor methods adopt a similar strategy to LoRA for initializing the update matrices $A$ and $B$, certain approaches, such as PiSSA diverge from this pattern by employing distinct initialization techniques for the adapter layers. Table 1 presents a comparison between LoRA, PiSSA, and the proposed ProRA method regarding the initialization of low-rank adapters $A$ and $B$. LoRA initializes its adapters randomly, whereas PiSSA employs singular vectors derived from the original weight matrices for initialization. In contrast, ProRA utilizes an orthonormal projection of $W$ to initialize its adapters. From a computational perspective, PiSSA necessitates performing singular value decomposition (SVD) for each layer, whereas ProRA only requires QR decomposition of $W$, which is significantly more efficient for large weight matrices commonly encountered in LLMs. Additionally, ProRA ensures that updates to the adapters remain orthogonal, unlike LoRA and PiSSA, where the adapters are not orthogonal to the frozen weights. This property inproves faster convergence during initial training steps.

Table 1: Comparison of PEFT Methods: LoRA, PiSSA, and ProRA.

| Method | LoRA | PiSSA | ProRA |
|---|---|---|---|
| Initialization | random | singular vectors | orthonormal projection |
| Complexity | Low | High (due to SVD) | Low |
| Orthogonality of Updates at initialization | × | × | ✓ |

## 4 EXPERIMENTS

We employed widely used language generation models (LLaMA2-7B (Touvron et al., 2023), Mistral-7B (Jiang et al., 2023), Gemma-7B (Team et al., 2024)) alongside an encoder-only Vision Transformer (ViT-B/16) (Dosovitskiy et al., 2020) model, pretrained on ImageNet. We validated our claims of improved initialization and faster convergence by testing the proposed ProRA on large-scale models and a diverse range of datasets (12 language and 3 vision task). The experiments were conducted on Nvidia A100-SXM4 (40GB) GPUs with a learning rate ranging from 1e-4 to 5e-5. For the rest of the experimental setup, we followed similar experimental setup as (Meng et al., 2024), using the AdamW optimizer and a batch size of 128. More details on experimental setup are provided in the Appendix A.3 and to ensure reproducibility of the ProRA, codes are also provided in supplementary material.

### 4.1 EVALUATION ON NATURAL LANGUAGE GENERATION (NLG) TASKS: WITH DIFFERENT LORA INITIALIZATION

We begin by comparing ProRA with different adapter initialization methods, namely PiSSA, LoRA, and full-parameter fine-tuning, on natural language generation (NLG) tasks. We tested our proposed

ProRA approach on a range of language generation tasks. All experiments were conducted using a 100K-sample subset and trained for a single epoch to minimize training time and resource usage. For math reasoning, we fine-tuned three models LLaMA 2-7B, Mistral-7B-v0.1, and Gemma-7B, on the MetaMathQA-40K (Yu et al., 2023) dataset and evaluated them on the GSM8K (Cobbe et al., 2021) and MATH (Hendrycks et al., 2021) validation sets. For code generation, the models were evaluated on the HumanEval (Chen et al., 2021) and MBPP (Austin et al., 2021) benchmarks. Based on the results in Table 7 ProRA achieves consistent improvements over most of the NLG tasks. Specifically, on LLaMA 2-7B, ProRA achieves the best performance across all tasks, outperforming PiSSA by up to 36.4% on HumanEval and 29.9% on MATH, and showing significant improvement compared to LoRA on some benchmarks. On Mistral-7B, ProRA delivers the strongest result on HumanEval (+8.4% over PiSSA) and matches the performance with leading methods on other tasks.

Table 2: Comparison of ProRA, PiSSA and LoRA on NLG tasks, with results averaged over three runs and reported with standard deviations.

| Model | Strategy | GSM8K | MATH | HumanEval | MBPP |
|---|---|---|---|---|---|
| LLaMA 2-7B | Full FT | 49.13±0.21 | 7.29±0.22 | 21.20±0.30 | 35.59±0.25 |
| | LoRA(gaussian) | 42.85±0.12 | 5.50±0.33 | 18.35±0.31 | 35.50±0.14 |
| | LoRA(kaiming) | 43.23±0.64 | 5.90±0.16 | 18.21±0.45 | 35.47±0.37 |
| | PiSSA | 53.22±0.55 | 7.47±0.34 | 21.92±0.38 | 37.24±0.63 |
| | **ProRA(ours)** | **55.59±0.17** | **9.7±0.09** | **29.9±0.48** | **40.1±0.43** |
| Mistral-7B | Full FT | 69.91±0.25 | 18.64±0.35 | 45.31±0.14 | 51.46±0.13 |
| | LoRA(gaussian) | 69.50±0.42 | 19.93±0.44 | 45.78±0.11 | 58.46±0.27 |
| | LoRA(kaiming) | 69.40±0.25 | 19.99±0.44 | 43.74±0.14 | 58.39±0.42 |
| | PiSSA | 73.31±0.23 | **23.12±0.52** | 46.88±0.25 | 62.55±0.58 |
| | **ProRA(ours)** | **72.72±0.44** | 22.4±0.49 | **50.8±0.74** | **62.73±0.37** |
| Gemma-7B | Full FT | 72.09±0.32 | 22.71±0.34 | 47.02±0.27 | 55.67±0.60 |
| | LoRA(gaussian) | 75.11±0.64 | 30.44±0.16 | 53.70±0.25 | 65.58±0.29 |
| | LoRA(kaiming) | 74.53±0.47 | 29.90±0.16 | 53.12±0.27 | 65.25±0.29 |
| | PiSSA | 77.78±0.32 | **31.33±0.33** | **54.31±0.28** | 66.17±0.43 |
| | **ProRA(ours)** | **78.11±0.27** | 27.9±0.19 | 50.4±0.75 | **66.3±1.01** |

As seen in Figure 2, on the MetaMath dataset, ProRA achieves the fastest reduction in training loss during the first 100 steps and later on, outperforming both LoRA and PiSSA. This suggests that ProRA learns more effectively right from the beginning of training. In Figure 3, ProRA also shows the highest gradient norm among the methods early in training. This indicates that ProRA enables larger and more expressive updates, specially at initial updates. Together, these trends show that ProRA is more responsive and adaptive in the initial phase of training compared to LoRA and PiSSA, which likely contributes to its stronger overall performance.

## 4.2 EXPERIMENTS OVER COMMONSENSE REASONING TASKS: WITH DIFFERENT LORA VARIANTS

We have also evaluated ProRA on eight commonsense reasoning benchmarks: BoolQ (Clark et al., a), PIQA (Bisk et al., 2020), SIQA (Sap et al., 2019), HellaSwag (HS) (Zellers et al., 2019), Winogrande (WG) (Sakaguchi et al., 2021), ARC-easy/challenge (Clark et al., b) and OpenBookQA (OBQA) (Mihaylov et al.). In Table 3, ProRA outperforms full fine-tuning (Gemma-2B) on most commonsense reasoning benchmarks. We conducted this study across several LoRA variants, and ProRA outperformed LoRA on 7 out of 8 datasets at rank 32, while also demonstrating superior performance on all datasets at the higher rank of 128.

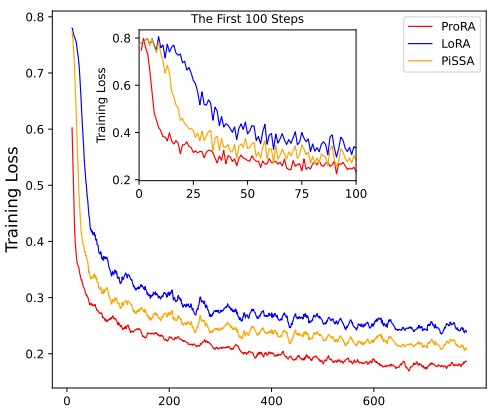 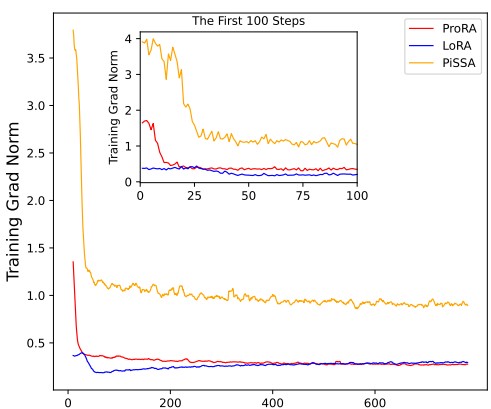

Figure 2: Training metrics for the MetaMathQA dataset: loss over training steps.

Figure 3: Training metrics for the MetaMathQA dataset: gradient over training steps.

Table 3: Commonsense Reasoning benchmarks using GEMMA-2B. Results are reported as accuracy (%) across various datasets.

| Method | BoolQ | PIQA | SIQA | HS | WG | ARC-e | ARC-c | OBQA |
|---|---|---|---|---|---|---|---|---|
| Full-FT | 63.57 | 74.1 | 65.86 | 70 | 61.95 | 75.36 | 59.72 | 69 |
| LoRA(r=32) | 63.11 | 73.44 | 63.2 | 47.79 | **52.95** | 74.78 | 57.16 | **67** |
| **ProRA (r=32)** | **64.49** | **80.35** | **72.97** | **89.63** | 51.22 | **77.39** | **58.7** | 61 |
| DoRA(r=1) | 62.17 | 68.77 | 55.93 | 32.95 | 51.22 | 68.81 | 48.72 | 55.6 |
| VeRA(r=2048) | 62.11 | 64.31 | 49.18 | 32 | 50.74 | 58.08 | 42.83 | 42.6 |
| SVFT | 62.26 | 70.18 | 56.7 | 32.47 | 47.04 | 69.31 | 50.08 | 58.4 |
| LoRA(r=128) | 66.06 | 80.36 | 74.56 | 91.36 | 53.04 | 79.25 | 59.73 | 65.2 |
| PiSSA(r=128) | 67.29 | 80.57 | 76.4 | 91.85 | 50.82 | 78.15 | 59.04 | 65.4 |
| **ProRA(r=128)** | **67.75** | **82.2** | **76.92** | **92.31** | **53.74** | **80.68** | **60.49** | **65.6** |

## 4.3 EVALUATION ON IMAGE CLASSIFICATION TASKS

For vision tasks, we evaluate our proposed ProRA method on three datasets: CIFAR-100 (Krizhevsky et al., 2009), RESISC45 (Ullah et al., 2022), and Flowers102 (Nilsback and Zisserman, 2008). We apply ProRA during fine-tuning under standard image classification settings. We evaluated ProRA on vision tasks using ViT-B as the backbone. As shown in the Table 4, ProRA achieves substantial improvements over other methods on all three datasets, outperforming LoRA, DoRA, and SVFT by large margins, most notably by +4.0% on CIFAR100 and over +18% on Resisc45, while provides comparable performance on Flowers102 datasets.

Table 4: Performance on vision classification tasks using ViT-B backbone. ProRA achieves superior performance while using fewer parameters than Full-FT. #Params is parameter count.

| Model | Method | #Params | CIFAR100 | Flower102 | Resisc45 |
|---|---|---|---|---|---|
| ViT-B | Full-FT | 85.8M | 85.35 | 98.37 | 68.03 |
| | LoRA (r=8) | 1.32M | 84.41 | 99.23 | 76.86 |
| | DoRA (r=8) | 1.41M | 85.03 | **99.30** | 76.95 |
| | SVFT (d=8) | 0.94M | 85.69 | 98.88 | 70.41 |
| | **ProRA(r=8)** | 1.32M | **89.70** | 99.07 | **95.04** |

## 4.4 ABLATION STUDY

Here, we present the ablation study conducted for ProRA. The ProRA method have two key aspects: the choice of relative ranks for low-rank updates and the selection of the projection matrix $P$. In this subsection, we discuss the impact of each aspect separately. We have also provided effect of ProRA on different transformer component and an empirical view of reduced geometric complexity of residual matrix in Appendix A.2

### 4.4.1 EVALUATION OF PRORA ON DIFFERENT RANKS

We conducted an ablation study on the proposed ProRA method by varying the adapter rank ($r = 8, 32, 128$). This analysis was performed on the commonsense reasoning benchmark using the Gemma-2B model, with results reported as the mean and standard deviation over three runs. As summarized in Table 5, the findings indicate a consistent trend: increasing the adapter rank leads to improved average performance across all evaluated datasets.

Table 5: Ablation on commonsense reasoning benchmarks using Gemma-2B with different ProRA ranks. Results reported as the mean and standard deviation over three runs

| Rank | BoolQ | PIQA | SIQA | HS | WG | ARC-e | ARC-c | OBQA |
|------|-------|------|------|-----|-----|-------|-------|------|
| 8 | 62.32±0.07 | 74.57±0.12 | 65.23±0.09 | 71.20±0.14 | 50.87±0.14 | 70.05±0.11 | 49.37±0.11 | 49.27±0.25 |
| 32 | 64.58±0.06 | 80.39±0.09 | 73.18±0.17 | 89.62±0.03 | 51.09±0.10 | 77.48±0.10 | 58.78±0.18 | 61.00±0.20 |
| 128 | **67.43±0.24** | **82.67±1.03** | **76.23±0.71** | **92.47±0.17** | **53.29±2.17** | **80.80±0.86** | **60.15±0.86** | **65.93±0.25** |

### 4.4.2 EFFECT OF DIFFERENT CHOICES FOR THE PROJECTION MATRIX

We investigated the effect of different choices for the projection matrix $P$, as presented in Table 6, by evaluating two construction strategies: randomized and deterministic. In the randomized approach, the orthonormal subspace is derived by projection of the weight matrix onto a random matrix, whose entries are independently sampled from a Gaussian distribution. We conducted experiments using both strategies on the LLaMA 2-7B model across GSM8K, MATH, HumanEval, and MBPP datasets. As shown in Table 6, when employing a randomized projection matrix, the proposed ProRA method outperforms LoRA, achieving up to a 3.41% increase in accuracy on the GSM8K dataset, with more modest gains observed on the other datasets. However, with the deterministic projection matrix constructed from the orthonormal subspace of the pretrained weight matrix, ProRA not only surpasses LoRA but also the randomized variant of ProRA, achieving a notable margin of improvement on all evaluated datasets. All experiments were conducted in a single run, and the corresponding results are reported in Table 6.

Table 6: Ablation on different choice of projection matrix $P$, on GSM8K, MATH, HumanEval, and MBPP using LLaMA 2-7B. ProRA$^R$ denotes the random projection matrix and ProRA$^D$ represents the deterministic projection matrix. Accuracy scores are reported for each task.

| Model | Method | GSM8K | MATH | HumanEval | MBPP |
|-------|--------|-------|------|-----------|------|
| LLaMA 2-7B | LoRA | 42.17 | 6.12 | 22.0 | 37.8 |
| | ProRA$^R$ | 45.48 | 6.38 | 22.3 | 38.4 |
| | **ProRA$^D$** | **55.72** | **9.8** | **25.6** | **39.7** |

## 5 CONCLUSION

In this work, we proposed Projection Aware Low-Rank Adaptation (ProRA), a unified framework that introduces low-rank adaptation along orthonormal directions while explicitly minimizing geometric complexity. By projecting pretrained weights onto orthonormal subspaces, ProRA not only enables structured and stable initialization but also preserves norm and gradient flow, leading to faster and more stable convergence. Our theoretical analysis and empirical results confirm that ProRA effectively reduces the geometric complexity of frozen residual components, which facilitates better generalization to downstream tasks.

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

# A APPENDIX

## A.1 THEORETICAL PROPERTIES OF PRORA

**Theorem 2.** The residual matrix $W_{\mathrm{res}}$ has lower geometric complexity than the original pretrained weight matrix $W$.

*Proof.* The squared Frobenius norm of $W$ is defined as

$$\|W\|_F^2 = \mathrm{tr}(W^\top W).$$

Substituting $W = W_{\mathrm{res}} + W_{\mathrm{proj}}$, we have

$$\|W\|_F^2 = \mathrm{tr}\left((W_{\mathrm{res}} + W_{\mathrm{proj}})^\top (W_{\mathrm{res}} + W_{\mathrm{proj}})\right)$$
$$= \mathrm{tr}(W_{\mathrm{res}}^\top W_{\mathrm{res}} + W_{\mathrm{res}}^\top W_{\mathrm{proj}} + W_{\mathrm{proj}}^\top W_{\mathrm{res}} + W_{\mathrm{proj}}^\top W_{\mathrm{proj}}).$$

Using orthogonality, $\mathrm{tr}(W_{\mathrm{res}}^\top W_{\mathrm{proj}}) = \mathrm{tr}(W_{\mathrm{proj}}^\top W_{\mathrm{res}}) = 0$, this simplifies to

$$\|W\|_F^2 = \|W_{\mathrm{res}}\|_F^2 + \|W_{\mathrm{proj}}\|_F^2.$$

Since both terms are non-negative, it follows that

$$\|W_{\mathrm{res}}\|_F^2 \leq \|W\|_F^2.$$

Hence, geometric complexity is measured by the squared Frobenius norm (as in Dirichlet energy) (Dherin et al., 2022), we have

$$\mathrm{GC}(W_{\mathrm{res}}) \leq \mathrm{GC}(W).$$

$\square$

## A.2 ABLATION STUDY

### A.2.1 EFFECT OF PRORA DIFFERENT TRANSFORMER COMPONENTS

Figure 4 and 5 investigates the impact of fine-tuning specific transformer components, including Query, Key, Value, Output, Up, Gate, and Down projections. The findings indicate that Query and Key have the smallest effect, followed by Value and Down, while Gate, Output, and Up have the greatest influence. This aligns with their functional roles: Query and Key mainly contribute to attention scoring, whereas the other components directly affect the transformation and retention of learned representations.

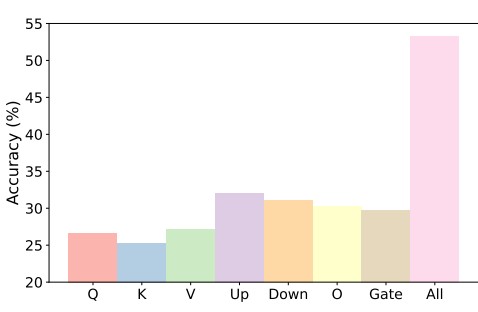 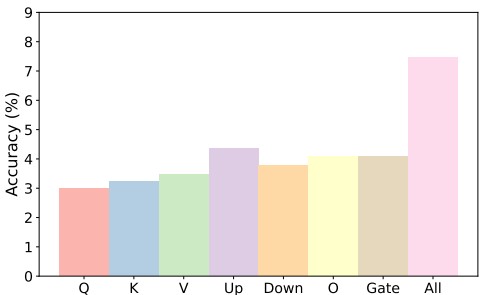

Figure 4: Effect of ProRA on different transformer component on GSM8K dataset (LLaMA-7B)

Figure 5: Effect of ProRA on different transformer component on GSM8K dataset (LLaMA-7B)

#### A.2.2 ANALYSIS ON GEOMETRIC COMPLEXITY ($W_{\text{RES}}$ Vs $W$)

Figure 6 presents an empirical analysis of reduced Geometric complexity of residual matrix from original weight matrix W. we have studied the geometric complexity at initial layers of the LLaMA 2-7B during training on MetaMathQA dataset, and find out that residual matrix have lower geometric complexity.

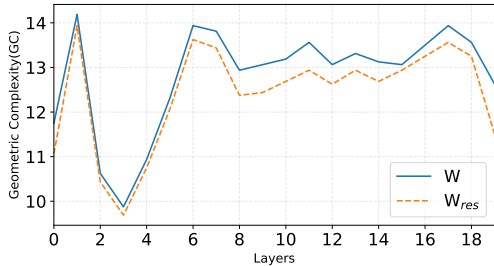

Figure 6: Analysis on Geometric Complexity ($W_{\text{res}}$ Vs $W$), during training MetaMathQA dataset on LLaMA 2-7B model.

### A.3 HYPERPARAMETER SETTING FOR DIFFERENT TASKS

In this section we will provide additional settings in order to reproduce our results. we have conducted our study on natural language and vision tasks.

#### A.3.1 NATURAL LANGUAGE GENERATION (NLG) TASK

We tested our proposed ProRA approach on a range of language generation tasks using LLaMA2-7B (Touvron et al., 2023), Mistral-7B-v0.1 (Jiang et al., 2023), Gemma-7B (Team et al., 2024), as discussed in Table 2 of main paper. In the Table 7 optimal learning rate for each model for the proposed approach ProRA. For NLG task we have utilised GSM8K (Cobbe et al., 2021), MATH (Hendrycks et al., 2021), HumanEval (Chen et al., 2021) and MBPP (Austin et al., 2021) benchmarks.

Table 7: Learning rate of ProRA on NLG tasks.

| Model | GSM8K | MATH | HumanEval | MBPP |
|---|---|---|---|---|
| LLaMA 2-7B | 1e-4 | 1e-4 | 1e-4 | 1e-4 |
| Mistral-7B | 5e-5 | 5e-5 | 2e-5 | 2e-5 |
| Gemma-7B | 3e-5 | 3e-5 | 2e-5 | 2e-5 |

### A.3.2 COMMONSENSE REASONING

All hyperparameter values used in our experiments for are listed in Table 8. LR represents Learning rate. We use the Hugging Face Transformers[1] and PEFT[2] libraries, which also provide access to training and evaluation datasets.

Table 8: Hyperparameters used for fine-tuning Gemma-2B on the Commonsense-15K dataset for ProRA.

| Hyperparameter | Value |
|---|---|
| Optimizer | AdamW |
| Learning rate | 3e-5 |
| Warmup steps | 100 |
| Batch size (train/eval) | 16 / 16 |
| Number of epochs | 50 |
| Weight decay | 0.0 |
| LR scheduler | Cosine |
| ProRA rank ($r$) | 8 |
| $\alpha$ | 8 |

### A.3.3 VISION TRANSFORMER

We fine-tune a pretrained ViT-B model on each vision dataset using ProRA, following a fixed set of hyperparameters (Table 9). ProRA is trained for 10 epochs on CIFAR-100 (Krizhevsky et al., 2009) and RESISC45 (Ullah et al., 2022), and for 30 epochs on Flower102 (Nilsback and Zisserman, 2008) to allow for better convergence. For other methods, we report results directly from their original implementations. We use the same Transformers and PEFT libraries as in the commonsense setup.

Table 9: Hyperparameters used for fine-tuning of ViT-B using proposed PEFT technique ProRA.

| Hyperparameter | Value |
|---|---|
| Model | ViT-B/16 |
| Batch size (train/eval) | 64 / 64 |
| Learning rate | 5e-4 |
| Weight decay | 0.01 |
| Warmup steps | 500 |
| LR scheduler | Cosine |
| ProRA rank ($r$) | 8 |
| $\alpha$ | 8 |
| ProRA target modules | query, key, value, dense |

---

[1] https://github.com/huggingface/transformers
[2] https://github.com/huggingface/peft

