# OpenReview forum: "ProRA: Projection Aware Low-Rank Adaptation for Parameter Efficient Fine-Tuning"
_ICLR.cc/2026/Conference — ICLR 2026 Conference Withdrawn Submission_

### Official Review · Reviewer_C9NV · 2025-10-27

**Soundness:** 3
**Presentation:** 2
**Contribution:** 2
**Rating:** 4
**Confidence:** 2

**Summary:**

This paper introduces ProRA (Projection Aware Low-Rank Adaptation), a parameter-efficient fine-tuning method for large language models. Unlike LoRA, which initializes adapters with random Gaussian noise and zeros, ProRA decomposes pretrained weights W into a trainable projected component W_proj and a frozen residual component W_res using orthonormal projection. The key innovation is initializing adapters A and B through orthonormal subspace projection (A = WU^T_{[:,:r]}, B = U_{[:,:r]}), which the authors claim enables faster convergence and reduces geometric complexity in frozen weights. The method is evaluated on NLG tasks (GSM8K, MATH, HumanEval, MBPP), commonsense reasoning benchmarks, and vision tasks, showing improvements over LoRA and comparable methods like PiSSA.

**Strengths:**

1. The paper provides an interesting connection between low-rank adaptation and geometric complexity (Dirichlet energy), offering theoretical justification that the residual matrix W_res has lower geometric complexity than the original weights.
2. The evaluation spans multiple modalities (language and vision), diverse tasks, and multiple model architectures (LLaMA2, Mistral, Gemma, ViT), demonstrating broad applicability.

**Weaknesses:**

1. The core idea of using structured initialization from pretrained weights is very similar to PiSSA. While ProRA uses orthonormal projection instead of SVD, the conceptual difference is incremental. The paper doesn't clearly articulate why orthonormal projection is fundamentally superior beyond computational cost.
2. While Theorem 2 proves ||W_res||²_F ≤ ||W||²_F mathematically, this is trivial from the Pythagorean theorem given orthogonality (Theorem 1). The connection to actual fine-tuning performance is not rigorously established—lower geometric complexity in frozen weights doesn't necessarily translate to better adaptation or generalization.
3. Insufficient ablation studies: No comparison of QR vs SVD initialization quality, Limited analysis of why orthonormal projection helps beyond norm preservation.

**Questions:**

1. Why does ProRA underperform on some tasks? Can you provide insight into why ProRA shows worse performance than LoRA on MATH and HumanEval with Gemma-7B, despite theoretical advantages? Are there task characteristics that make orthonormal initialization less suitable?
2. Can you provide more direct empirical evidence linking the reduced geometric complexity of W_res to improved fine-tuning outcomes?  For instance, do tasks where ProRA excels correlate with larger reductions in geometric complexity?
3. Table 6 shows deterministic projection outperforms random, but how does QR-based orthonormal projection compare to SVD-based initialization (PiSSA) when both use the same rank?  What specific advantages does orthonormality provide beyond what principal components offer?

---

### Official Review · Reviewer_rxfL · 2025-10-28

**Soundness:** 2
**Presentation:** 2
**Contribution:** 1
**Rating:** 2
**Confidence:** 4

**Summary:**

The paper proposes Projection Aware Low-Rank Adaptation (ProRA), a parameter-efficient fine-tuning framework that initializes LoRA adapters through orthonormal projections of pretrained weights. The method aims to preserve the geometric structure of pretrained models, reduce the geometric complexity of the frozen components, and achieve faster convergence and improved transfer performance.

**Strengths:**

- This paper provides a novel perspective on the initialization of LoRA from the spectral initialization.


 - The statement in this paper is easy to follow.

 - The experimental section is extensive and well-designed.

**Weaknesses:**

- W1. The two challenges identified in this paper are not well supported, which makes the claims appear unconvincing. Compared with full fine-tuning, LoRA—despite having far fewer trainable parameters—often adapts to new tasks efficiently. If the authors wish to argue that LoRA exhibits slow convergence at the beginning of training, they should provide direct comparisons with full fine-tuning, either theoretically or empirically (e.g., loss curves or gradient norms in early training stages). Furthermore, the claim that higher geometric complexity hinders effective adaptation during fine-tuning should be substantiated with explicit literature citations or empirical validation.

- W2. The paper claims to propose a novel initialization method based on spectral decomposition. However, non-zero initialization has already been extensively studied in prior works ([1], [2], [3]). Specifically, [1] points out that spectral initialization purely derived from pretrained weights can cause fine-tuning failure, yet this paper still fine-tunes along the top-r subspaces of the pretrained model without noting that issue. [2] provides a rigorous theoretical analysis comparing zero and non-zero initializations, offering solid guarantees that are absent in this paper. Moreover, [3] also introduces a distinct non-zero initialization strategy, but the authors did not include it in their experimental comparisons.


 - W3. The paper claims that the frozen part in ProRA exhibits lower geometric complexity than that of the original LoRA, thereby leading to better transfer of knowledge. However, this argument is not well supported. The connection between reduced geometric complexity for the frzeon part and improved transfer performance is asserted without sufficient theoretical justification or empirical validation. The authors should clearly explain why and how lowering geometric complexity contributes to better adaptation.


[1] Xu Z, Min H, MacDonald L E, et al. Understanding the Learning Dynamics of LoRA: A Gradient Flow Perspective on Low-Rank Adaptation in Matrix Factorization[J]. arXiv preprint arXiv:2503.06982, 2025.

[2] Li S, Luo X, Tang X, et al. Beyond Zero Initialization: Investigating the Impact of Non-Zero Initialization on LoRA Fine-Tuning Dynamics[J]. arXiv preprint arXiv:2505.23194, 2025.

[3]Wang S, Yu L, Li J. Lora-ga: Low-rank adaptation with gradient approximation[J]. Advances in Neural Information Processing Systems, 2024, 37: 54905-54931.

**Questions:**

- q1. In Table 2, line 351 (Mistral-7B section), the highlighted result seems incorrect — PiSSA is 73.31 over ProRA 72.72 on GSM8K.

 - q2. In Table 3, the number of decimal places is inconsistent across datasets (e.g., some results are shown with two decimals, others with none). Could the authors standardize the reported precision—for example, using two decimal places throughout—to ensure readability and consistency across all comparisons?

---

### Official Review · Reviewer_63jh · 2025-10-30

**Soundness:** 2
**Presentation:** 1
**Contribution:** 2
**Rating:** 2
**Confidence:** 4

**Summary:**

This paper studies ProRA, which adopts geometric complexity as a theoretical lens to explain and improve PEFT. Specifically, ProRA initializes the adapter matrices by projecting the original weight matrix onto its orthonormal subspace, while keeping the residual weight matrix frozen. Both theoretical analyses and empirical evaluations demonstrate the effectiveness of the proposed method.

**Strengths:**

* This paper studies an interesting and relevant topic, as PEFT is crucial for adapting foundation models.
* This paper provides detailed theoretical analysis regarding their design.

**Weaknesses:**

* The motivation is not clear. Why do random and zero initializations lead to slow convergence and limit expressiveness? I don’t think the two key challenges are clearly articulated. Could you provide a more detailed example illustrating how the updates in the original LoRA lead to more informative gradients and longer training durations? In my experience, the gradient norm remains reasonable during the initial training phases. Moreover, I cannot grasp the intuition behind high geometric complexity, how is the high complexity of LLMs related to the challenges faced by LoRA?
* Simplifying the geometric complexity to mappings within a single linear layer seems too coarse. Moreover, in Appendix A.2.2, the authors provide some analysis of geometric complexity during training. Is this analysis computed based on Equation (1) or Equation (2)?
* The comparison in Table 2 is incomplete. The authors only compare with LoRA and PiSSA, while there are numerous recent LoRA-like baselines, such as DoRA and VeRA (as shown in Table 3), as well as other variants including AdaLoRA, LoRA-GA, LoRA-Pro, and LoRA-Rite.
* The notation is inconsistent. Future versions should align the use of $W_{res}$ and $W_{\text{res}}$.
* There are also issues with the boldface in Table 2. The best GSM8K score for Mistral-7B is achieved by PiSSA, not ProRA. The overall improvement over PiSSA is limited for both Mistral-7B and Gemma-7B.
* Some details regarding the learning rate are missing. Could you also provide the learning rate settings used for LoRA and ProRA?

**Questions:**

See questions in the weaknesses part.

---

### Official Review · Reviewer_7fof · 2025-11-03

**Soundness:** 3
**Presentation:** 3
**Contribution:** 3
**Rating:** 4
**Confidence:** 4

**Summary:**

This paper proposes Projection-Aware Low-Rank Adaptation (ProRA), a PEFT method that orthogonally projects pretrained weights into a low-rank subspace and initializes LoRA adapters using the projected components. The residual component remains frozen and, by construction, has lower geometric complexity. The authors theoretically show orthogonality and reduced Frobenius norm of frozen weights, and empirically validate ProRA across a wide range of LLMs (LLaMA2/Mistral/Gemma) and ViTs, demonstrating faster convergence and improved downstream performance over LoRA, PiSSA, and other baselines.

**Strengths:**

The core idea ： projection with orthonormal basis + residual freezing —is elegant and easy to integrate into existing PEFT pipelines；

The work leverages geometric complexity (Dirichlet energy) to justify freezing the residual；

Gains on GSM8K, HumanEval, MBPP, commonsense benchmarks；

Demonstration of larger initial gradients and faster loss reduction confirms the argument that ProRA leverages pretrained structure more efficiently than random initialization:

**Weaknesses:**

The authors argue QR is cheaper than SVD, but no runtime/memory benchmarks are provided；

Although the paper compares ProRA with LoRA and some of its variants, the experimental evaluation does not yet cover the full spectrum of state-of-the-art PEFT approaches. In particular, several recent methods based on orthogonal constraints, directional decomposition, or SVD-style parameterization (e.g., DoRA, NoRA, FLoRA) are highly relevant in terms of architectural motivation and geometric regularization. The absence of comparisons with these representative techniques may limit a comprehensive assessment of ProRA’s novelty and advantages within the current PEFT landscape.

**Questions:**

same as above

---

### Note · Authors · 2025-11-24

I have read and agree with the venue's withdrawal policy on behalf of myself and my co-authors.